# Selective Antifungal Activity and Fungal Biofilm Inhibition of Tryptophan Center Symmetrical Short Peptide

**DOI:** 10.3390/ijms22158231

**Published:** 2021-07-30

**Authors:** Shuli Chou, Qiuke Li, Hua Wu, Jinze Li, Yung-Fu Chang, Lu Shang, Jiawei Li, Zhihua Wang, Anshan Shan

**Affiliations:** 1Institute of Animal Nutrition, Northeast Agricultural University, Harbin 150030, China; choushli@mail.sysu.edu.cn (S.C.); 15636103001@163.com (Q.L.); Wh0411592x@163.com (H.W.); 18104476918@163.com (J.L.); sshanglu@163.com (L.S.); lijiawi@163.com (J.L.); jingjingkuyihui@163.com (Z.W.); 2Department of Population Medicine and Diagnostic Sciences, College of Veterinary Medicine, Cornell University, Ithaca, NY 14853, USA; yc42@cornell.edu

**Keywords:** Tryptophan center symmetrical short peptide, fungus-targeted activity and biofilm inhibition, low drug resistance and side-effects

## Abstract

*Candida albicans*, an opportunistic fungus, causes dental caries and contributes to mucosal bacterial dysbiosis leading to a second infection. Furthermore, *C.**albicans* forms biofilms that are resistant to medicinal treatment. To make matters worse, antifungal resistance has spread (albeit slowly) in this species. Thus, it has been imperative to develop novel, antifungal drug compounds. Herein, a peptide was engineered with the sequence of RRFSFWFSFRR-NH2; this was named P19. This novel peptide has been observed to exert disruptive effects on fungal cell membrane physiology. Our results showed that P19 displayed high binding affinity to lipopolysaccharides (LPS), lipoteichoic acids (LTA) and the plasma membrane phosphatidylinositol (PI), phosphatidylserine (PS), cardiolipin, and phosphatidylglycerol (PG), further indicating that the molecular mechanism of P19 was not associated with the receptor recognition, but rather related to competitive interaction with the plasma membrane. In addition, compared with fluconazole and amphotericin B, P19 has been shown to have a lower potential for resistance selection than established antifungal agents.

## 1. Introduction

Approximately 25% (about 1.7 billion) of the worldwide population at any one time suffers from superficial fungal infections of skin, and nails [1], and an unprecedented rise in the rate of emergence of antifungal-resistant strains has been observed. Hence, *C. albicans* cgmcc 2.2086 compromises fecal microbiota transplantation (FMT), which is effective in treating recurrent *Clostridioides difficile* infection (CDI) [2]. Furthermore, *C. albicans* was also confirmed to promote tooth decay by altering microbial ecology [3]. However, antifungal resistance generates less public attention than viral and bacterial infections [4,5]. Indeed, compared to other pathogens, the eukaryotic-like nature of fungal cells, which possess substantial similarities to human cells has limited the scope of fungi-specific drug discovery and development [6]. Additionally, the current broad-spectrum antifungal drugs which may destroy common microbes and allow unwanted microbes to flourish even if they show low toxicity to human cells results in a change of balance in the microbes of areas of the body such as the vagina, intestine and mouth, and leading to adverse clinical consequences [7]. New therapeutic options with low resistance potential and discriminatory efficacy to control emerging fungal infections are a priority to resolve these problems.

A group of small antimicrobial peptides (AMPs) with multiple functions, including wound healing, antibacterial activity, anti-biofilm, and anti-cancer activity with modifications of their primary sequences [8,9,10,11,12], have recently shown great promise as antifungal agents. In addition, they kill pathogens through non-specific membrane permeabilization, which displayed a low propensity to induce resistance [13,14], therefore representing an attractive candidate as medical coatings and therapeutic drugs [15,16]. Biofilms generated by pathogenic bacteria and some eukaryotes like *C. albicans* are hardy and difficult to remove [17]. These growth formations are made of organisms typically fastened onto tissue surfaces via a variety of adhesins. Biofilms are also characterized by a wide range of biomolecules that render them resistant to antimicrobial agents. However, many AMPs have come to show promise as antimicrobial agents, particularly against *C. albicans*, which has been implicated as a widespread pathogen in human dental caries. Previous studies have reported that their rapid mode of action, unlike conventional antifungal drugs that may target fungal sterols or enzyme receptor sites, can potentially reduce treatment duration, making AMPs excellent antifungal drug candidates [18]. In addition, some reports have indicated that screening antimicrobial peptides libraries presents a perfect opportunity for uncovering novel potent and selective targeting motifs for a specific target [19,20,21]. This study implemented a search of sequences filtered from the Antimicrobial Peptide Database (APD). Derived and synthetical peptides were eliminated from this search (http://aps.unmc.edu/AP/) (Schedule 1). As the main active parameter, the hydrophobic residues Leu (L), Phe (F), and Ile (I), which occurred more frequently in antifungal peptides, were chosen as the primary components of novel antifungal short peptide candidates for testing. Additionally, our previous researches have shown that central-symmetric α-helix sequence contributed to improving activity and selectivity [22,23]. Therefore, sequences distributed into that short central-symmetric α-helical template (++ hyh W hyh ++), (h, hydrophobic amino acid; +, cationic amino acid; y, basic amino acid) were regarded as potential targeted antibiotic alternatives, which not only decrease the production cost with short sequences but also reduce the likelihood of drug resistance section with possible non-receptor site via a membrane permeabilization mechanism. Hydrophilic residues Ala(A), Cys(C), Ser(S), which occur more frequently in antifungal peptides, were chosen to modify the whole hydrophobicity of the sequences. Finally, five short peptides with being amidated at their C-terminus were synthesized and evaluated here (Scheme 1). The minimum inhibitory concentrations of peptides against bacteria and fungi were first assessed to confirm the spectrum of antibacterial activity. Time-killing, bacterial survival counts, human erythrocyte cytotoxicity, human embryonic kidney 293T cells, and pig kidney cells were also assessed to demonstrate the antimicrobial rate cell selectivity. In addition, salt, acid sensitivity assays, and drug resistance of peptide candidates against fungi were tested to indicate antimicrobial activity in various conditions. Fluorescent spectrography, whole-cell ELISA, protein-lipid overlay assays, scanning electron microscopy (SEM), transmission electron microscopy (TEM), and confocal laser scanning microscopy (CLSM) assays were employed to investigate the membrane-disruption mechanism of the potential peptides synthesized in our study. The overall objective of this study was to facilitate the development of peptide-based specific agents capable of permeabilizing plasma membranes of target fungal cell lines.

## 2. Results

The Matrix-Assisted Laser Desorption/Ionization Time of Flight Mass Spectrometry (MALDI-TOF MS) (Figure 1) and Reverse-Phase High-Performance Liquid Chromatography (RP-HPLC) results suggested that all the peptides were successfully synthesized, with their measured molecular weights being close to the theoretical molecular weights (Table 1) and the purities of the peptides were more than 95% (Figure 2).

### 2.1. Antimicrobial Activity

The antimicrobial activities of the peptides and control antifungals against Gram-negative bacteria, Gram-positive bacteria and fungi are shown in Table 2 and Table 3. To confirm the potential of fungal antibiotic replacement, amphotericin B and fluconazole antimicrobial experimental groups were also tested simultaneously. It was suggested that all of these five designed peptides composed of APD-filtered amino acids were effective on fungi cells. As shown in Table 2, P17 and P19 exhibited antimicrobial specificity and efficiency with 1–4 μM MIC value against tested fungi, including fluconazole-resistant *C. albicans* 56214, while with higher MIC value against tested bacteria. But P17 still showed some antibacterial activates, including the probiotic *L. rhamnosus* 1.0385 at slightly above antifungal MIC values. Thus, P19 was chosen as an ideal peptide to analyze further.

### 2.2. Structure Variability of the Peptides

CD spectroscopy was employed to determine the secondary structures of the peptides in mimicking physiological environment (PBS) and mimicking the microbial membrane environment (sodium lauryl sulfate (SDS) and trifluoroethanol (TFE). The spectra of the peptides in 10 mM PBS, 50% TFE and 30 mM SDS were shown in Figure 3. In the mimicking membrane environment, P1, P2, and P3 showed α-helical structure, with CD spectra showing minimum peaks at 208 nm and 220 nm. In contrast, in the physiological environment, P1, P2, P3 tended to unordered conformations with a minimum peak at approximately 198 nm. P17 showed β-sheet structure, with CD spectra showing a minimum peak in the 200–220 nm range, and a positive peak in the 195–200 nm range in the different mimicking environments. P19 showed β-sheet structure with CD spectra showing a minimum peak in the approximately 210 nm range. A positive peak in the minimum 198 nm mimicking membrane environment, while P19 showed unordered conformation in the physiological environment.

Time killing curve and fractional cell survival were employed to measure fraction of cell survival of *C. albicans* cgmcc 2.2086 upon treatment at the ½ MIC, MIC, and 2 MIC levels of peptides at various exposure times determine the sterilization speed of P19. Figure 4a showed that P19 killed *C. albicans* cgmcc 2.2086 in 60 s and less than 5 s at MIC values and 2 MIC values, respectively. One hour or more was required to eliminate 90% *C. albicans* cgmcc 2.2086 at the ½ MIC level, suggesting that the killing rate of P19 was tightly correlated with concentration.

The fractional survival rates of cells treated with P19 at 2 MIC values of *E. coli*.

ATCC 25922, *S. aureus* ATCC 29213, *L. rhamnosus* 1.0385 and *C. albicans* cgmcc 2.2086 were measured to confirm the inhibited effect on the growth of bacteria and fungi. Figure 4b showed that P19 caused some inhibition on the growth of *E. coli* ATCC 25922, *S. aureus* ATCC 29213 and *L. rhamnosus* 1.0385, while it induced much higher inhibitory levels against *C. albicans* cgmcc 2.2086.

Biofilms formed by *C. albicans* are generally significantly more resistant to antimicrobial treatment than living cells and are thus harder to eradicate. Here, the activity of P19 against biofilms was tested. Results indicated that P19 could induce obvious biofilm eradication (60%) at 4 MIC levels (Figure 5) and it caused a similar level of antimicrobial activity against *C. albicans* biofilms with amphotericin B at 16 MIC value.

### 2.3. Biocompatibility of the Peptides

Biocompatibility of the designed peptides, which is a prerequisite for clinical application, was investigated. To demonstrated the toxic effects of peptides on normal cells, human red blood cells cytotoxicity was tested on kidney 293T cells and pig kidneysfrom the same lines (Figure 6). The activities of all designed peptides against fresh, healthy human erythrocytes, human embryonic kidney 293T and pig kidney PK cells were shown as Figure 6. All the designed peptides induced less than 5% hemolysis at their MIC values, especially for P19, which showed less than 5% hemolysis even at the highest tested concentration. The peptide cytotoxicity against 293T cells and PK cells also confirmed that the cell viabilities after exposure to these peptides remained 80% or greater even at their highest tested concentrations. The tissue cell lines treated with P19 retained viabilities of 100% at most of the concentrations tested.

### 2.4. Sensitivity upon Salt and Acid Condition

The susceptibility of bioactive agents to the physiological environment was another significant effect that could prevent them from achieving clinical application. Thus, the antimicrobial activity of the potential peptide P19 against *C. albicans* cgmcc 2.2086 in the presence of physiological concentrations of different salts or acidic environments was further evaluated (Table 4). The result showed that P19 retained its MIC values at 4 μM in the presence of most salts, except for Na^+^ and Ca^2+^, which increased MIC values to 32 μM. Additionally, P19 showed compromised activity with MIC of 4 μM in acidic conditions.

### 2.5. Drug-Resistance

The potential development of drug resistance to *C. albicans* cgmcc 2.2086 induced by P19 and antifungals was tested using repeated assays in the presence of sub-MICs of the antimicrobial agent. Figure 7 showed that a 20-passage repeating treatment with fluconazole or amphotericin B at sub-lethal concentration resulted in a 1024-fold increase in MIC value compared with the first passage. In contrast, the rise in peptide levels was accompanied by a 2-fold MIC increase over40 days only, suggesting that resistance to P19 was not acquired easily.

### 2.6. Membrane Permeabilization and Integrity

Disc3-5 and PI were used to detect the membrane permeability of *C. albicans* cgmcc 2.2086 treated with P19, melittin, fluconazole, or amphotericin B. Figure 8a showed the plasma membrane depolarization experiment, and the fluorescence intensity represents the results. The fluorescence intensity of P19 was the highest. P19 induced a more significant change of membrane depolarization compared with other tested agents. In contrast, minor change of membrane depolarization occurred under treatment with fluconazole or amphotericin B. Figure 8b showed the flow cytometry experiment. The results visually showed the distribution of cells by the distribution of PI dye markers, and the areas where cells were concentrated represent the state of cells as live or die. P19 and melittin induced more than 90% increase of PI fluorescent signal while the application of antifungals induced none.

To visually observe morphology, ultrastructure and integrity of membrane of *C. albicans* cgmcc 2.2086 treated with P19, SEM, TEM and CLSM were employed. Figure 9a,b showed that the cell membrane surface treated with P19 was broken and roughen while untreated cells had integrated and smooth surfaces. Figure 9c,d showed that P19 induced outflow of intracellular content. The results indicated that P19 exerted its antifungal activity by compromising the integrity of the cell membrane (Figure 10).

### 2.7. Membrane Binding Affinity

To investigate the binding affinity of P19 to the cell membrane and cellular phospholipids, whole-cell ELISA, and protein-lipid overlay assays using biotin-labeled P19 and HRP-Streptavidin were employed. Figure 11 showed that the binding affinity of P19 to *C. albicans* cgmcc 2.2086 was significantly higher than to *S. aureus* ATCC 29213 and *L. rhamnosus* 1.0385 (*P* < 0.05), and P19 showed just a little higher but nonsignificant binding affinity to *C. albicans* cgmcc 2.2086 (*P* > 0.05). Figure 12 further showed that P19 recognized the negatively charged phospholipids including phosphatidylserine (PS), phosphatidylglycerol (PG), phosphatidylinositol (PI), cardiolipin, PtdIns, PtdIns(4)P, PtdIns(3, 5)P_2_ and PtdIns(4, 5)P_2_. Figure 13 showed a high binding affinity of P19 with LPS and LTA, which might inhibit the binding of the peptide with phospholipids in bacteria.

## 3. Discussion

In this study, we developed a class of APD-filtered central-symmetric AMPs with five sequences were synthesized. The high MIC values of P1, P2, P3, P17, P19 against tested fungi indicated that this short central-symmetric template of 11-residues had bacteria-killing activity and was also capable of fungicidal action. We also confirmed that hydrophobic amino acids (L, F, and I) and hydrophilic amino acids.(A, S and C) as the most frequent residues found in natural antifungal peptides. In addition, the antimicrobial spectrum of P1, P2, and P3 demonstrated that basic amino acids showed only a slight effect on the antibacterial spectrum. Additionally, the more specific antimicrobial activity of P19 suggested that lower hydrophobic amino acid might be a benefit for improving the antimicrobial selectivity, which was associated with that the high hydrophobic amino acid, Leu-rich peptides, could indiscriminately insert or penetrate more deeply into negatively charged phospholipids. And this specific activity of phenylalanine-replacement substituted residues may be due to the presence of a bulky benzene ring inducing steric hindrance, changing the secondary structure of the peptide, a key factor for antimicrobial activity [23], which was supported by the secondary structures of these five peptides that P1-P3 showed α-helical structure in mimicking environments but P17 and P19 showed β-sheet structure in membrane mimicking environments.

Additionally, the CD results of these five peptides also suggested that not only a non-ordered structure in PBS but transformed to an α-helical structure in the membrane-mimetic environment could induce higher broad antimicrobial activity, a non-ordered structure in PBS. Still, it transformed to a β-sheet structure that could improve the selective antimicrobial activity against fungi. Our microbial survival counts result showed that P19 could inhibit all gram-negative bacteria, gram-positive bacteria, and fungi cells with varying degrees. This also indicated that selective antimicrobial activity of P19 might be via an affinity for charges and hydrophobic sites located on cell membrane rather than via a receptor-mediated pathway [24,25,26]. The results of the drug resistance trial also support this view.

The primary limitations for an in vivo application of peptide-based biomaterials stem from possible toxic effects on normal cells and sensitivity to the physiological environment [27,28]. Especially for this study because of the similarity of fungal cells and mammalian cells. Many reports have indicated that increased length of the peptides could facilitate rapid peptide anchor and insert into the lipid bilayer [29], which might induce higher cytotoxicity and lead to lower sensitivity to degradation [14]. Furthermore, it is known that salts compromise the binding activity of membrane-active peptides by disrupting electrostatic interactions between the peptides and any potential cell membrane receptors [30]. These novel peptides showed higher than 80% cell viability even at the highest concentration of 128 μM. The MIC values of P19 in the presence of physiological salt concentrations increased somewhat were a little raised, especially in the presence of Na^+^. This is primarily arising due to the high ionic strength, which can interfere with the electrostatic attraction between cationic peptides and anionic bacterial lipid bilayers [31,32,33]. And this is a common phenomenon for cationic peptides. Thus, these two results demonstrated that this fungal-specific 11-residues peptide was a safe and cost-effective antimicrobial agent and suggested exerting antimicrobial activity via bacterial membrane permeabilization as previously reported broad-spectrum antimicrobial agents.

To further validate the antimicrobial mechanism of P19, fluorescence spectroscopy and electron microscopy were performed using melittin, fluconazole, and amphotericin B as controls. Membrane depolarization and flow cytometry results showed that only membrane-active P19 and melittin increased the fluorescence and enabled the entry of the PI probe, which indicated a characteristic of peptide-membrane interactions. Combining with the apparent damage of *C. albicans* membrane surface observed by SEM, leakage of the intracellular contents detected by TEM and localization of the FITC-labeled P19 detected by CLSM, the action mechanism of P19 was suggested to occur in permeabilization of the plasma membrane and leakage of cytoplasm, which was the main reason for little alteration in the presence of drug-resistant variants.

The anti-biofilm and fast sterilization activity of P19 suggested that selective antimicrobial activity of P19 was not only for the lower hydrophobic amino acid. Then, to further check the selective activity of P19 between fungi and other bacteria based on the peptide-membrane interaction mechanism, the affinity of peptides to *E. coli*, *S. aureus*, *L. rhamnosus* and *C. albicans* were further detected using whole-cell binding assay. Results showed that a given level of P19 had a higher binding affinity with *C. albicans* than with bacterial samples of the same density. Still, the nonsignificant binding affinity between *C. albicans* and *E. coli* suggested that P19 showed precisely antimicrobial activity against fungi without specific binding. Negatively-charged lipopolysaccharides (LPS) or lipoteichoic acids (LTA), which are present in large amounts in the cell envelopes of Gram-positive and Gram-negative bacteria, may interact with peptides via charge residue, forming a barrier to reduce the concentration of peptides around the plasma membranes, thereby blunting their antimicrobial effects. Then, the high binding ability of peptide P19 with LTA and LPS indicated that the selective antibacterial activity of P19 might be associated with the competitive combination of P19 with LPS (LTA) and plasma membrane [34]. In addition, since this potential peptide showed membrane disruption discriminately without receptor-mediated pathway, lipid binding and membrane permeabilization are key components of the mechanisms of action of many such antimicrobial agents [35]. We then focused on interactions between P19 and lipid and analyzed the selective antimicrobial activity based on the lipid membrane composition. Figure 9 has identified PIPs as lipid targets, such as PtdIns(4)P, PtdIns(3, 5)P_2,_ and PtdIns(4, 5)P_2_ on the plasma membrane, to which P19 caused membrane disruption, bleb formation, and ultimately cell lysis. In addition to PIPs, P19 also showed binding affinity to other phospholipids such as PI and PS, which were common components of the fungal plasma membrane (more) and animal cell membrane (less) but are largely absent from bacteria [36,37,38]. Thus, the composition of the phospholipids of the fungal plasma membrane that enriched in PI and PS without negative-charged LPS and LTA was a benefit for increasing concentration or accumulation on typical fungal cell surface membranes of P19, and then improve the peptide bioactivity functions to disrupt fungal cell membrane integrity, resulting in fungal tissue injury or death. In addition, P19 did show somewhat binding ability with phosphatidylglycerol (PG) and cardiolipin, two main components of Gram-negative and Gram-positive bacterial plasma membranes, which give P19 at least some antibacterial capabilities, although bacterial generally showed more excellent resistance to P19 antimicrobial activity than fungi when peptide concentrations and target cell densities were normalized, and this was in accordance with the microbial survival rates results. It was suggested that P19 showed selective antimicrobial activity based on the various compositions of lipids on the plasma membrane. Previous reports have shown that high levels of negatively-charged phospholipids [39,40], increased membrane surface area [41] and strongly expressed phosphatidylserine (PS) in tumor cells [42] may serve as high-affinity binding targets for short peptides such as P19, indicating that this short peptide may have some utility as an anti-tumor agent.

## 4. Materials and Methods

### 4.1. Bacteria and Fungi

*Escherichia coli* (*E. coli*) ATCC 25,922, *Staphylococcus aureus* (*S. aureus*) ATCC 29,213 and *S. aureus* ATCC 25,923 were provided by the College of Veterinary Medicine, Northeast Agricultural University (Harbin, China), and *E. coli* UB1005 was kindly provided by the State Key Laboratory of Microbial Technology (Shandong University, China). *Lactobacillus rhamnosus* 7469, *Lactobacillus rhamnosus* 1.0911, *Lactobacillus rhamnosus* 1.0385, *Lactobacillus rhamnosus* 1.0925 and *Streptococcus thermophiles* (*S. thermophilus*) YM-C were obtained from the Key Laboratory of Food College, Northeast Agricultural University. *C. albicans* cgmcc 2.2086*, C. tropocalis* cgmcc 2.1975*, C. parapsilosis* cgmcc 23*,*989 were purchased from the China General Microbiological Culture Collection Center (Beijing, China). Clinical isolated *C. albicans* SP3903, *C. albicans* SP3937*, C. albicans* SP3902 and *C. albicans* isolated from alveolar fluid were obtained from the Medical College of Nanchang University. Fluconazole-resistant *C. albicans* 56,214 was provided by the Zhongshan Hospital of Fudan University.

### 4.2. Peptide Synthesis

The filtered result was analyzed by R programming. The helical wheel projection was performed online (http://rzlab.ucr.edu/scripts/wheel/wheel.cg) (Schedule 1B). The peptides designed in this study were synthesized by the Sangon Biotech (Shanghai, China), and the actual molecular weights were tested by matrix-assisted laser desorption/ionization time-off light mass spectrometry (MALDI-TOF MS; Linear Scientific Inc., USA). The purity of the peptides (95%) was assessed by reverse-phase high-performance liquid chromatography (HPLC) using column of Shimdzu Inertsil ODS-SP column (4.6 × 250 mm, 214 nm, 20 μL) and a non-linear water/acetonitrile gradient containing 0.1% trifluoroacetic at a flow rate of 1.0 mL min^−1^. The general schematic and three-dimensional structure projection of the potential peptide was graphed by using ChemDraw V10.0 or predicted online (http://zhanglab.ccmb.med.umich.edu/I-TASSER/).

### 4.3. Circular Dichroism (CD) Measurements

CD spectra of 150 µM peptides were measured at room temperature in 10 mM phosphate-buffered saline (PBS) (mimicking an aqueous environment), 50% TFE (mimicking the hydrophobic environment of the microbial membrane), and 30 mM SDS micelles (negatively charged prokaryotic membrane comparable environment) on a J-820 spectropolarimeter (Jasco; Tokyo, Japan), with a quartz cuvette with a 1.0-mm path length. The spectra were recorded from 195-250 nm at a scanning speed of 10 nm/min, and an average of 3 scans was collected for each peptide. The acquired CD spectra were then converted to the mean residue ellipticity with the following formula:θ_M_ = (θ_obs_·1000)/(c·l·n)(1)
where θ_M_ is the mean residue ellipticity [(deg·cm^2^)/dmol], θ_obs_ is the observed ellipticity corrected for the buffer at a given wavelength [md, eg], c is the peptide concentration [mM], l is the path length [mm], and n is the number of amino acids.

### 4.4. Antimicrobial Assays

The peptide antimicrobial activity was determined according to the Clinical and Laboratory Standards Institute method, with modifications. Bacteria concentration was measured by ultraviolet spectrophotometer, and fungi concentration was confirmed by McMillan’s turbidimetric method, which compared the fungi solution with the standard McFarland tube in front of the horizontal line on the paper with the naked eye. Because the light has different refract in different liquids. The standard solution is composed of sulfuric acid and barium chloride, which is equivalent to 1*10^8^ CFU fungi. After cultivation, the growth of fungi and bacteria was measured by TECAN automatic enzyme label instrument. Briefly, bacterial cells were cultured in MHB until logarithmic phase, yeast colonies cultured in Yeast Peptone Dextrose Agar (YPDA) were picked and diluted in RPMI 1640 growth medium buffered with morpholine propane sulfonic acid (MOPS), and then 50 μL of bacterial or fungi cell solution (0.5–1 × 10^4^ CFU/mL) was added to each well of sterile 96-well plates. Each well contained 50 μL of AMPs dissolved in BSA; the final concentrations of the peptides ranged from 0.5 to 128 μM. The lowest concentration of peptide with no microbial growth being observed were measured after incubation at 37 °C for 24 h for bacteria or at 28 °C for 48 h for fungi. Broth with microbial cells and un-inoculated broth were used as positive control and negative controls, respectively.

The time-kill kinetics of the potential peptide for *C. albicans* cgmcc 2.2086 was further investigated by analyzing the fractional cell survival after various peptide exposure times to determine the killing time curve. Briefly, the microbial cells were treated with a peptide at a 1× MIC concentration. At various periods (0, 5, 10, 30, 60, 300, 600, 3000 and 6000 s), microbial suspensions were diluted 10- and 100-folds, and then plated on solid medium plates using 50 μL diluted suspension. Microbial colonies were counted after 28 °C for 48 h of incubation.

Microbial survival counts of the potential against four typical microbial strains were also made to confirm the differences in antimicrobial efficacy of peptide constructs on bacterial and fungal targets. The microbial cells were treated with peptides at a 1× MIC concentration for 10 min. The subsequent steps were consistent with the time-killing kinetics method (bacterial colonies were formed and counted after 37 °C for 24 h (bacteria) or 28 °C for 48 h (fungi) of incubation. All the above tests were performed at least three times.

Biofilm eradication assay was further tested to confirm the activity of P19, which was monitored using the microtiter plate assay shown as Paola Saporito et al. described [43] with growth media as RPMI 1640 growth medium buffered with MOPS and biofilm incubating for 48 h.

### 4.5. Cytotoxicity Assays

The cytotoxicity of the peptides was determined with three cell types, including the Pig kidney cells, HEK 293T cells and fresh, healthy human red blood cells (donated by Zhihua Wang) via the MTT dye reduction assay and hemolysis assay [44].

### 4.6. Salt and Acid Sensitivity Assays

The salt and acid sensitivities of the peptides were measured using with modified method to test MIC values; tested peptides were incubated in the presence of different final concentrations of physiological salts. The subsequent steps were consistent with the MIC determination protocol [26].

### 4.7. Drug-Resistance

Resistance development of *C. albicans* cgmcc 2.2086 against antifungals was explored using a sequential passaging method, described in our previous study [22]. Briefly, MIC testing was first conducted for P19, fluconazole and amphotericin B. After a 24 h incubation, fungal cells growing in a well with a half-MIC level were harvested and diluted to 0.5–1 × 10^4^ CFU/mL using RPMI 1640 growth medium buffered with morpholine propane sulfonic acid (MOPS). The inoculum was subjected to subsequent passage MIC testing, and the process was repeated for 30 days. The fold change in MIC was plotted against the number of passages.

### 4.8. Membrane Permeabilization and Integrity Assays

#### 4.8.1. Cytoplasmic Membrane Depolarization Assay

To analyze membrane disturbances due to P19, melittin, fluconazole or amphotericin B-treated, cells (1 × 10^4^ cells/mL) of *C. albicans* cgmcc 2.2086 were incubated with antimicrobial agents at their respective MICs for 1 h at room temperature; the cells were then harvested by centrifugation and resuspended in 1 mL PBS (pH 7.4). Cytoplasmic membrane potential was observed with 0.4 μM DiSC_3_-5 (Sigma Chemical Co., USA). Changes in fluorescence were measured from 0 to 800 s with a F-4500 fluorescence spectrophotometer (Hitachi, Japan) at an excitation wavelength of 622 nm and an emission wavelength of 670 nm.

#### 4.8.2. Flow Cytometer Assay

Cell membrane permeabilization after treatment with P19, melittin, fluconazole or amphotericin B were also detected via the propidium iodide influx assay. In brief, *C. albicans* cgmcc 2.2086 cells (1 × 10^4^ cells/mL) were treated with antifungal agents for 1 h at 28 °C, then washed, harvested by centrifugation and resuspended in PBS. The cells were treated with 10 μg/mL propidium iodide and incubated for 30 min at room temperature. The uptake of propidium iodide into these cells was analyzed with a FACS flow cytometer (Becton–Dickinson, San Jose, CA) with a laser excitation wavelength of 488 nm. The flow cytometry results can be interpreted as follows. Region Q1 was identified as a living cell region; it represents cells treated with PBS. Q2 region represents the dead cell region. Ignored the surrounding adherent scattered points, the most concentrated scattered points represented the largest cell density. The areas where cells were concentrated represent the state of cells as live or die.

#### 4.8.3. SEM, TEM and CLSM Characterization

The direct visualization of *C. albicans* cgmcc 2.2086 membrane and the specimens treated with peptide or not were examined using a HITACHI S-4800 SEM, HITACHI H-7650 TEM and Leica TCS SP2 CLSM. For sample preparation, fungal cells were incubated for 1 h at 28 °C with P19 at 1× MIC or in the absence of peptide as a control. After incubation, the cells were harvested by centrifugation at 5000× *g* for 5 min, washed three times with PBS, then processed as previously described [9].

### 4.9. Membrane Binding Affinity Assays

#### 4.9.1. Whole Cell ELISA

Whole-cell ELISA assay was performed to compare the binding affinity of P19 with Gram-negative, Gram-positive bacteria, and fungal cells. Briefly, 100 μL 1 × 10^7^ CFU/mL microbial cells resuspended in carbonate buffer solution (CBS, Ph = 9.5). The suspension was added to wells on a 96-well plate for coating overnight at 37 °C, followed by fixing by ethanol and air-drying. Plates were blocked with gelatin for 2 h, and then the plate was washed with PBST for 4 times and incubated with 100 uL of PBST containing biotin-labeled peptides at a final concentration of 16 μM for 1 h at 37 °C. Th503-504en, incubation with 1:20,000 dilution of HRP-Streptavidin for 15 min at 37 °C after washing with PBST 4 times. Finally, the absorbance of OD_450_ was determined by using a spectrophotometer (Infinite 200 pro, Tecan, China) followed by washing the wells 4 times using the Tetramethylbenzidine membrane peroxidase (TMB) system.

#### 4.9.2. Protein–Lipid Overlay Assay

Membrane-bound P19 to various phospholipids (including Membrane Strips and PIP Strips) (Echelon Biosciences) was measured according to the manufacturer’s instructions. Briefly, the membrane strips were blocked with gelatin for 1 h at 37 °C, the plate was washed with PBST 4 times and incubated with 100 uL of PBST containing biotin-labeled peptides at a final concentration of 16 μM for 1 h at 37 °C. Plates were incubated with 1:20,000 dilution of HRP-Streptavidin for 15 min at 37 °C after washing with PBST 4 times. Peptide binding was detected by ECL as P19 labeled with biotin.

### 4.10. Binding Affinity to LPS or LTA

The peptide binding affinity to LPS or LTA was dependent on the probe bound to cell-free LPS or LTA. Briefly, 50 μg/mL LPS from *E. coli* or 50 μg/mL LTA from *S.aureus* and 5 μg/mL BC were kept in the dark at room temperature for 4 h. Then, peptides were added to the mixtures at minimum inhibitory concentrations and cultured for another 1 h at room temperature. Fluorescence was measured (excitation λ = 580 nm, emission λ = 20 nm, Excitation bandwidth = 9 nm, Emission bandwidth = 20 nm, *Z*-axis height = 20,000 μm) on a Spectro fluorophotometer (the Infinite 200 pro, Tecan, China). 10 μg mL^−1^ polymyxin B as control.

### 4.11. Statistical Analysis

All data were subjected to a one-way analysis of variance (ANOVA) and significant differences between the means were evaluated by Tukey’s test for multiple comparisons. The data were analyzed and diagrammed by using the GraphPad Prism 5.0. Quantitative data were expressed as the mean ± standard error (SE).

## 5. Conclusions

In summary, short sequence peptides with sequences derived using APD-filters showed efficient and rapid selective activity against fungi, tolerance to a broad range of physiological conditions, high biocompatibility and low-likelihood drug-resistance, suggest ideal candidates for drugs against *C. albicans* induced infection. Furthermore, selective mechanism and change of structure of the peptides indicated that fungicidal peptide P19 had targeting antimicrobial activity through membrane attraction and properties of the peptide instead of receptor targeting. Short peptides with specific sequences can be tailored to target specific cell surface lipids on pathogens, allowing researchers to develop antimicrobials with great precision and a low potential for developing target cell resistance, which facilitates the development of further development novel non-receptor targeted antimicrobial agents.

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
