# Peer review of "Selective Antifungal Activity and Fungal Biofilm Inhibition of Tryptophan Center Symmetrical Short Peptide"

_ijms, 2021, doi:10.3390/ijms22158231_

Round 1
Reviewer 1 Report
The details of my revision has been carried out in the attached pdf file.

Reviewer 2 Report
The manuscript #ijms-1303089, entitled “Selective Antifungal Activity and Fungal Biofilm Inhibition of 2 Tryptophan Center Symmetrical Short Peptide” by Chou et al. presents the synthesis, as well as antimicrobial activities of short peptides, with emphasis on pathogenic Candida albicans fungus. The presented paper is quite well written, and brings novelty to the field. The design of the study is mostly logical and well planned. Most of my comments concern minor (mostly editorial) issues.
Major issue
Out of the tested peptides, P19 was chosen for further studies. The justification included herein states that “Based on this, P19 exhibited the best antimicrobial specificity (…)” (Line 100). In my opinion, P17 displays comparable activities during initial antifungal screening (Table 3). Thus, the authors should elaborate more on the fact why P19 was chosen over P17.
Minor issues
Line 32: Clostridium difficile should be italicized.
Line 71: The authors should elaborate why only kidney cell lines were chosen for investigation.
Line 96: Remove “:” form the title.
Lines 126 and 130 (and throught manuscript): please insert the symbol “½” instead of 1/2.
Line 211: ATCC 29213 instead of “2,213”.
Line 279: S. aureus should be italicized.
Line 318: Italics.
Line 319: ATCC 25923 instead of “ATCC 25,923”.
Line 340: space between the 20 and µL.
Line 350: Please add the formula according to the journal guidelines.
Line 425: Should be “described”.
Line 452: Please provide more details, such as PMT voltage, Ex and Em slits etc.
Figures and tables
Tables 2 and 3. Could you please elaborate what “sof” stands for when the authors mention “MICsof”. Or if this is editioral issue (lack of space between s and of) please correct it.
Table 2. Could you elaborate if usage of anti-fungal drugs (AMB and FLC) was necessary in case of antibacterial screening. In my opinion, the lack of activity is a desirable effect.
Table 3. In the table the abbrev. “cgmcc” is lacking italics, wheres throughout the manuscript each time the authors mention this strain, the italics is present. In my opinion, if “cgmcc” stands for “China General Microbiological Culture Collection Center” it should not be italicized. Please correct this throught the manuscript or justify why italicizing is necessary herein.
Table 4: Could you discuss why in the presence of NaCl the activity of P19 is reduced? What is the hypothesis on molecular level.
Figure 3: Please expand the abbrev. “PBS”, “SDS”, and “TFE” in the caption.
Figure 4: Microbial names in the panel b should be italicized.
Figure 5: Please provide the strain symbol, next to C. albicans in the caption. Are the results statistically significant?
Figure 6: Provide the cell lines in the caption. Additionally, can you discuss why P3 and P17 display heamolytic activities, whereas no cytoxicity? In my opinion it is not suprising since erythrocytes are generally recognized as much more vulnerable towards stresses than cultured cells. However, it should be discussed.
Figure 7: The result is very interesting and, in my opinion, it should be highlighted in the abstract. Additionally, the strain, which acquired 2-fold MIC resistance should be investigated in comparison towards non-treated initial cells to answer what are the exact, possible mechanisms of resistance towards P19.
Figure 8: Please provide the strains, next to C. albicans in the caption.
Figure 9: Similarly, add “cgmcc” in the caption.
Figure 11: Microbial names should be italicized.
